# Expanding diversity of tick-borne phleboviruses (*Phlebovirus mukawaense*, Mudanjiang phlebovirus, Gomselga Virus, and Onega tick phlebovirus) in Russia

Mikhail Y. Kartashov[1], Valentina Y. Kurushina[1], Alexey O. Yanshin[1,2], Alina S. Zheleznova[1], Kirill A. Svirin[1], Tatyana V. Tregubchak[1], Vladimir A. Ternovoi[1], Alexander P. Agafonov[1], Anastasia V. Gladysheva [1,2]*

1 State Research Center of Virology and Biotechnology "Vector", Kol'tsovo, Russia, 2 Physics Department, Novosibirsk State University, Novosibirsk, Russia

* gladysheva_av@vector.nsc.ru

## Abstract

Tick-borne phleboviruses represent emerging pathogens with zoonotic potential, yet their distribution across Asian Russia remains poorly characterized. This study investigated the prevalence, genetic diversity, and evolutionary dynamics of phleboviruses in 1,078 individual *Ixodes persulcatus* ticks collected from 143 locations across Asian Russia during summer 2023. Samples underwent PCR screening, high-throughput sequencing for genome reconstruction, phylogenetic analysis, and AlphaFold protein structural modeling. We detected 27 phleboviral isolates belonging to *Phlebovirus mukawaense* (MKWV), Mudanjiang phlebovirus, Onega tick phlebovirus, and Gomselga virus, with prevalence rates of 0.6%, 0.1%, 0.3%, and 1.5%, respectively. Phylogenetic analysis suggested Gomselga virus belongs within the MKWV species complex. Complete coding sequences of Russian MKWV isolates enabled high-confidence structural predictions for key proteins. The predicted tertiary structure of the MKWV nucleoprotein exhibits strong similarity to those of Rift Valley fever virus and Toscana virus, despite limited sequence identity. The MKWV nucleoprotein features a conserved globular core with a well-defined RNA-binding cleft. Segment-specific phylogenetic incongruence among Primorsky MKWV isolates indicated potential reassortment events. Additionally, coinfections involving MKWV and Alongshan virus, *Borrelia miyamotoi*, or *Rickettsia* spp. were identified. These findings significantly expand the known geographic range and genetic diversity of tick-borne phleboviruses in Russia. The integration of genomic and structural data provides a robust framework for functional annotation and highlights the evolutionary stability of essential viral domains. Further research should focus on virus isolation and investigating antigenic properties and replication capacity in mammalian cells.

**Data availability statement:** The nucleotide sequences of the L segment fragment obtained in this study have been deposited in the GenBank (https://www.ncbi.nlm.nih.gov/labs/virus) under the following accession numbers: PP476819-PP476824 and PV843499 for MKWV; PP476825-PP476840 for Gomselga virus; PP476841-PP476843 for OTPV; and PP476844 for MJPV. Additionally, complete coding sequences of MKWV have been deposited as follows: PP525061-PP525067 for the L segment, PP525068-PP525074 for the M segment, and PP525075-PP525081 for the S segment.

**Funding:** The work was supported by the Ministry of Science and Higher Education of the Russian Federation (The Federal scientific-technical program for genetic technologies development for 2019-2030, Agreement № 075-15-2025-526). The funders had no role in study design, data collection and analysis, decision to publish, or preparation of the manuscript.

**Competing interests:** The authors have declared that no competing interests exist.

## Introduction

Tick-borne viruses represent a significant public health concern in many regions of Russia, particularly in its Asian territories, where high incidence rates of tick-borne encephalitis virus (*Orthoflavivirus encephalitidis,* TBEV) and Ixodes tick-borne borreliosis are consistently reported. The ongoing epidemiological situation is not only characterized by the expanding prevalence of well-established tick-transmitted pathogens but also by the emergence and detection of novel agents whose clinical relevance and public health impact remain incompletely understood. Among these emerging pathogens are several recently identified tick-borne viruses associated with febrile illnesses in humans. Notable examples include Severe fever with thrombocytopenia syndrome virus (*Bandavirus dabieense*, SFTSV) [1–3], Bourbon virus (*Thogoto virus bourbonense*) [4,5], Alongshan virus, Jingmen tick virus [6–8], Yezo virus (*Orthonairovirus yezoense*) [9,10], Toyo virus (*Uukuvirus toyoense*) [11,12], and Songling virus (*Orthonairovirus songlingense*) [13]. These findings highlight the need for continued surveillance and research to assess the diversity, distribution, and potential pathogenicity of tick-associated viruses in endemic regions.

The ecosystems of southern Siberia are highly conducive to the persistence and circulation of diverse tick vectors and pathogens responsible for tick-borne infections [14]. The vast territory of southern Eastern Siberia borders or is in close proximity to regions in China where the majority of newly identified viruses associated with human febrile illnesses have been detected [15,16]. This geographical closeness enhances the potential for cross-regional transmission and local establishment of emerging tick-borne viruses, also increases the risk of recombination and the emergence of viruses with new properties, emphasizing the need for ongoing surveillance and epidemiological monitoring in this ecologically sensitive area.

The *Phlebovirus* genus within the *Phenuiviridae* family belongs to the broad ecological group of arthropod-borne viruses, which are transmitted from mammalian hosts through the bites of hematophagous arthropods. Most species within this genus are arthropod-associated viruses and can cause a range of diseases in humans and animals, including febrile illnesses, hemorrhagic fevers, and encephalitis. The phlebovirus genome consists of three segments of single-stranded negative-sense RNA (ssRNA(-)). The large (L) segment encodes the RNA-dependent RNA polymerase (RdRp); the medium (M) segment encodes the glycoproteins Gn and Gc, and in some viruses, also includes an additional non-structural protein; the small (S) segment encodes the nucleoprotein and, via an ambisense coding strategy in many members of the genus, a non-structural protein (NS), which often functions as a virulence factor by interfering with host immune responses [17,18].

Among the *Phlebovirus* genus members, Heartland virus (*Bandavirus heartlandense*, HRTV) and SFTSV have attracted particular attention due to their ability to cause severe clinical disease and high case fatality rates in infected humans. SFTSV was first identified in China in 2009 as a novel and highly pathogenic bunyavirus, posing a significant threat to human health. Since its discovery, SFTSV infections have been reported in several countries across East and Southeast Asia, including South Korea, Japan, Vietnam, and Pakistan [19–22]. HRTV was discovered in the

United States in 2011 and is associated with a clinical syndrome similar to SFTSV. Phylogenetic analyses reveal a high degree of genetic similarity between HRTV and SFTSV. However, HRTV is genetically distinct and represents a separate viral lineage within the *Phlebovirus* genus [23].

In 2018, a novel tick-borne virus Mukawa virus (*Phlebovirus mukawaense*, MKWV) was identified in *Ixodes persulcatus* ticks collected on Hokkaido Island, Japan. MKWV belongs to the *Phlebovirus* genus within the *Phenuiviridae* family. Serological investigations of wild animals inhabiting the region revealed the presence of neutralizing antibodies against MKWV in Yezo deer (*Cervus nippon yesoensis*) and raccoons (*Procyon lotor*), indicating exposure to the virus or his closest genetically related virus. These findings provide evidence for the existence of endemic foci of MKWV with zoonotic potential in Japan. Subsequent studies conducted in 2019 and 2021 reported the detection of MKWV in *Ixodes persulcatus* and *Haemaphysalis concinna* ticks in Heilongjiang Province, China. Furthermore, a genomic fragment of MKWV was also identified in *Dermacentor silvarum* ticks from northern regions of China, suggesting a broader geographical distribution and potential expansion of the virus beyond Japan into continental Asia.

A genetically related virus *Kuriyama virus* (KURV), also belonging to the *Phlebovirus* genus was first detected and isolated from *Ixodes persulcatus* ticks on Hokkaido, Japan [24]. Another novel phlebovirus species, Mudanjiang phlebovirus (MJPV), was identified through high-throughput next-generation sequencing (NGS) in *Ixodes persulcatus* ticks collected in Mudanjiang City, Heilongjiang Province, China [25]. Subsequently, MJPV-specific genomic sequences were detected in four additional tick species: *Dermacentor silvarum*, *Haemaphysalis longicornis*, *Rhipicephalus microplus*, and *Haemaphysalis concinna* [26], indicating a broader vector range and potential for wider geographic dissemination. *Onega tick phlebovirus* (OTPV) was detected in the taiga tick *Ixodes persulcatus* in northeastern Asia, including regions in Russia and China [25]. These findings highlight the increasing diversity of tick-borne phleboviruses in East Asia and underscore the importance of continued surveillance for emerging arboviruses in both vectors and wildlife.

To date, the available literature on the distribution and high tick specificity of phleboviruses in the European part of Russia remains limited [27,28]. For instance, *Gomselga virus* has been detected in *Ixodes persulcatus* ticks in Chelyabinsk Oblast, the Republic of Karelia, and in Asian Russia, specifically in the Tyva Republic. Recent phylogenetic and genomic analyses have demonstrated that previously described Gomselga viruses in Russia are closely related to MKWV and likely belong within the same viral species [29]. These findings suggest that what was historically classified as Gomselga virus represents part of the broader MKWV species complex, highlighting the extensive geographic distribution and significant genetic diversity of this group across both East Asia and Europe.

The present study focuses on the detection and molecular-genetic characterization of MKWV, including Gomselga-like virus variants, MJPV, and OTPV in southern Siberia, Russia. These findings contribute to a better understanding of the circulation patterns of emerging and potentially pathogenic phleboviruses in natural foci across the country and highlight the need for expanded surveillance of tick-borne viruses in understudied regions.

## Materials and methods

### Ethics statement

The study does not involve human or animal samples; therefore, no approval from any institutional ethics committee was sought.

Tick collection was performed exclusively from vegetation using the flagging method; no ticks were collected from humans seeking medical care or from animal hosts (mammals or birds). According to the legislation of the Russian Federation, this type of environmental sampling does not require approval from an ethical committee or specific collection permits, as it does not involve vertebrate animals or human subjects. Field work was conducted by qualified specialists from the zoological departments of the Regional Centers for Hygiene and Epidemiology (under the Federal Service for Surveillance on Consumer Rights Protection and Human Wellbeing, Rospotrebnadzor), who hold the necessary certifications and authorizations for entomological surveillance activities in accordance with Russian federal and regional regulations.

All collection sites were located on publicly accessible lands or areas where routine epidemiological monitoring is routinely performed by these authorized institutions.

## Tick Sample Collection

Tick collection was conducted by flagging vegetation across 143 locations within 12 administrative regions in Asian Russia. Sampling took place during the summer season of 2023 to coincide with peak tick activity. Collected specimens were morphologically identified to species level using established taxonomic keys [30,31]. All ticks collected for this study were adult specimens (imago). Prior to nucleic acid extraction, individual ticks were surface-sterilized by washing with 70% ethanol, followed by air-drying under sterile conditions to minimize external contamination. Tick samples were mechanically homogenized in 300 µL of sterile physiological saline using a TissueLyser LT homogenizer (Qiagen, Hilden, Germany). This instrument operates by high-speed shaking of sample tubes in the presence of solid particles, ensuring rapid and uniform disruption of the hard exoskeleton. Two steel beads (4 mm diameter; Servicebio, Wuhan, China) were added to each tube containing an individual tick specimen. Homogenization was performed at a shaking frequency of 50 Hz (50 shakes/sec) for 5 minutes.

## Nucleic acid extraction

Viral RNA was extracted from the homogenates using an ExtractRNA Kit (Evrogen, Moscow, Russia), according to the manufacturer's instructions. First-strand cDNA synthesis was carried out using the MMLV RT Kit (Evrogen, Moscow, Russia) with random hexamer primers. To monitor the efficiency of nucleic acid extraction, a control PCR assay targeting a fragment of the mitochondrial cytochrome c oxidase subunit I (*COI*) gene of *Ixodes persulcatus* was performed for all samples. Amplification of this endogenous host gene confirmed successful RNA/DNA preservation, adequate homogenization, and absence of potent PCR inhibitors, thereby validating the quality of the nucleic acid preparations prior to downstream molecular screening for phleboviruses. Nevertheless, the possibility that some samples contained intact tick DNA while viral RNA was degraded cannot be completely excluded.

## PCR testing

All tick samples (cDNA) were screened for the presence of phlebovirus genetic material using a conventional PCR assay targeting a fragment of the viral L segment. Amplification was performed with specific primers PhlP2 and PhlM2 [32] on a T-1000 Thermal Cycler (Bio-Rad, Hercules, CA, USA) in a 25 µL reaction volume. The PCR cycling conditions consisted of an initial polymerase activation step at 95 °C for 5 min, followed by 38 cycles of denaturation at 95 °C for 15 s, annealing at 50 °C for 20 s, and extension at 72 °C for 50 s, with a final elongation step at 72 °C for 4 min. To ensure reliability and prevent contamination, negative controls (nuclease-free water substituted for template) were included in each run alongside nucleic acid extraction controls. Amplified products corresponding to the expected fragment size of approximately 507 bp, representing partial sequences of the phlebovirus L segment, were separated by electrophoresis on 2% agarose gels stained with ethidium bromide and visualized under UV illumination. All positive amplicons were purified from agarose gels using the TIANquick Midi Purification Kit (TIANGEN, Beijing, China) according to the manufacturer's protocol. Sanger sequencing was performed using the BigDye Terminator v3.1 Cycle Sequencing Kit (Applied Biosystems, Waltham, MA, USA), and nucleotide sequences were determined via capillary electrophoresis on an automated sequencer ABI 3500/3500xl (Applied Biosystems, Waltham, MA, USA). The obtained sequence data were processed using SnapGene v.3.2.1 (GSL Biotech LLC, San Diego, CA, USA).

## Genome enrichment and NGS sequencing

For targeted PCR amplification of MKWV genomic variants intended for whole-genome sequencing, a fusion DNA polymerase Pfu-Sso7d (Biolabmix, Novosibirsk, Russia) was used in combination with a custom-designed panel of

oligonucleotide primers developed in this study (S1 Table). Library preparation for high-throughput sequencing was performed using the NEBNext Ultra II FS DNA Library Prep Kit for Illumina (NEB, Hitchin, UK), which involved enzymatic fragmentation, end repair, dA-tailing, and adapter ligation in a single reaction mix. Sequencing was carried out on an Illumina MiSeq platform (Illumina, San Diego, CA, USA).

Resulting FASTQ files were processed to remove adapter sequences, short reads, and low-quality sequences (Phred quality score < 20 and length < 30 nucleotides) using fastp v0.20.1 (https://github.com/OpenGene/fastp/) [33]. Quality-trimmed reads were then aligned to the Chinese reference genome of MKWV NE-TH3 strain (GenBank ID: ON408114, ON408115, ON408116) retrieved from NCBI GenBank using BWA-MEM v0.7.18 (https://github.com/lh3/bwa) [34]. SAM/BAM file processing and analysis were conducted using Samtools v1.11 [35]. Consensus sequences were extracted from the aligned BAM files using iVar v1.2.2 [36], with variants called at positions exhibiting ≥70% nucleotide identity.

## Sequence alignments and phylogenetic reconstruction

All available genome sequences were downloaded for phleboviruses from NCBI (https://www.ncbi.nlm.nih.gov/labs/virus), accessed on 1 July 2025 and their sequence information was recorded (S2 Table). Multiple sequence alignment was performed using MEGA X (PSU, Philadelphia, PA, USA). These programs were also used to calculate sequence identities. A custom Python script was developed using the Biopython library to compute pairwise sequence identity matrices. The analysis was performed using Python v3.12.7 with the following libraries: Biopython v1.85 for sequence parsing and manipulation, Pandas v2.2.3 for data handling, Seaborn v0.12.2, and Matplotlib v3.7.2 for visualization.

Maximum likelihood trees were constructed using MEGA X (PSU, Philadelphia, PA, USA). The evolutionary history was inferred by using the Kimura 2-parameter model. The tree with the highest log likelihood is shown. The percentage of trees in which the associated taxa clustered together is shown next to the branches. Initial tree for the heuristic search were obtained automatically by applying Neighbor-Join and BioNJ algorithms to a matrix of pairwise distances estimated using the Maximum Composite Likelihood approach, and then selecting the topology with superior log likelihood value. A discrete Gamma distribution was used to model evolutionary rate differences among sites (+G). The rate variation model allowed for some sites to be evolutionarily invariable (+I). The tree is drawn to scale, with branch lengths measured in the number of substitutions per site. Bootstrap support values (1000 replicates) were estimated at each node. The resulting phylogenetic tree was visualized using iTOL v7.1 (https://itol.embl.de/, accessed on 27 October 2025) [37].

## Statistical analysis

The minimum phleboviral infection rate was estimated using 95% confidence intervals (95% CI). The 95% CI was calculated using Wilson's estimator without correction for continuity (https://pedro.org.au/wp-content/uploads/CIcalculator.xls).

## Nucleotide sequence accession numbers

The nucleotide sequences of the L segment fragment (approximately 455 nucleotides in length, n = 27) obtained in this study have been deposited in the GenBank (https://www.ncbi.nlm.nih.gov/nucleotide) under the following accession numbers: PP476819-PP476824 and PV843499 for MKWV; PP476825-PP476840 for Gomselga virus; PP476841-PP476843 for OTPV; and PP476844 for MJPV. Additionally, complete coding sequences of MKWV (n = 7) have been deposited as follows: PP525061-PP525067 for the L segment, PP525068-PP525074 for the M segment, and PP525075-PP525081 for the S segment.

## Structural annotation of MKWV genome

Spatial structure models of proteins from the S, M, and L segments were generated using the AlphaFold 3 server (Google DeepMind, London, UK; https://alphafoldserver.com/welcome, accessed on 27 October 2025) [38]. The models were

validated against known structures from PDB using the FoldSeek server (Seoul National University, Seoul, South Korea; https://search.foldseek.com/, accessed on 27 October 2025) [39]. The search was performed in the PDB100 database using TM-score and sequence identity metrics.

Structural models were selected based on both the per-residue confidence coefficient (pLDDT, scaled from 0 to 100) and the predicted TM-score (pTM) provided by AlphaFold 3, where pLDDT estimates the expected deviation in Cα atom distances and pTM offers an overall measure of the predicted model's global accuracy. To compare model accuracy, pairwise structural alignments were performed between viral protein structures and AlphaFold-generated models using the Pairwise Structure Alignment tool from the RCSB Protein Data Bank (Piscataway, NJ, USA; https://www.rcsb.org/alignment/, accessed 27 October 2025), employing TM-align for alignment [40]. Topological similarity was evaluated using the root mean square deviation (RMSD) and TM-score, where values closer to 1 indicate a near-perfect structural match. Tertiary structures were visualized with UCSF ChimeraX v1.15rc (University of California, San Francisco, CA, USA) [41].

Pairwise identity alignment of secondary structures of viral proteins were generated with an FoldScript v1.2. (Institute for the Biology and Chemistry of Proteins, Lyon, France), https://foldscript.ibcp.fr/ (accessed on 27 October 2025) [42].

## Results

### Identification of phlebovirus sequence fragments in ticks

A total of 1,078 individual *Ixodes persulcatus* ticks were collected and analyzed in this study across five regions in Asian part of Russia: Irkutsk region, Tyva Republic, Zabaykalsky Krai (Chita region), Khakassia Republic, and Primorsky Krai (Table 1). All ticks were tested using PCR with pan-phleboviral primers [32]. A total of 27 positive *Ixodes persulcatus* ticks were detected. Screening for phlebovirus RNA revealed that the highest prevalence was observed in ticks from Irkutsk region (4.5%, 95% CI: 2.4–8.3) and the Republic of Tyva (3.8%, 95% CI:1.9–7.3), whereas the lowest prevalence was detected in Primorsky Krai (1.1%, 95% CI:0.4–2.8). The overall average minimum infection rate across the studied regions was 2.5% (95% CI:1.7–3.6). Typing of the identified phlebovirus variants based on a partial L segment sequence (455 bp) analysis demonstrated that Gomselga virus was the predominant virus, with an overall prevalence of 1.5% (95% CI:0.9–2.4), followed by Mukawa virus (MKWV) at 0.6% (95% CI:0.3–1.2). Onega tick phlebovirus (OTPV) was detected less frequently (0.3%, 95% CI:0.1–0.8), while Mudanjiang phlebovirus (MJPV) was found only as a single Primorye-29 isolate, corresponding to a minimum infection rate of 0.1% (95% CI:0.02–0.5).

### Phylogenetic relationship of identified phlebovirus sequence fragments

Maximum likelihood phylogenetic analysis of the fragment encoding the RNA-dependent RNA polymerase (RdRp) revealed that MKWV, including Gomselga variants, MJPV, and OTPV each form three distinct monophyletic clades (Fig 1). These well-supported clades are clearly separated from one another and from other related phleboviruses, confirming their genetic distinctiveness and consistent with their classification as separate viral lineages within the *Phlebovirus* genus.

Russian MKWV isolates exhibit 91−97% nucleotide sequence identity in the RdRp-encoding fragment with Chinese isolates, while showing lower similarity (89−91%) to the Japanese prototype MKW73 isolate (GenBank ID: NC043511). Among the Russian isolates, MKWV Primorye-49 isolate was the most divergent, sharing only 90−91% sequence identity with other Russian and Chinese MKWV variants and displaying 89% identity with the Japanese MKWV isolate (Fig 1). This distinct genetic profile of MKWV Primorye-49 isolate supports a potentially unique evolutionary trajectory of this variant within the MKWV complex circulating in southern Siberia and the Russian Far East.

At the same time, Russian Gomselga virus isolates exhibit a high degree of genetic similarity among themselves, ranging from 91% to 100% nucleotide identity in the RdRp-encoding fragment. In contrast, their sequence identity with

**Table 1. Distribution of *Ixodes persulcatus* ticks with detectable phleboviruses.**

| Russia region | No. Ticks examined | No. Positive ticks | Mukawa virus | Gomselga virus | Onega tick phlebovirus | Mudanjiang phlebovirus |
|---|---|---|---|---|---|---|
| | | Prevalence (95% CI) | | | | |
| Irkutsk region | 200 | 9 | 0 | 6 | 3 | 0 |
| | | 4.5% (2.4-8.3) | | 3.0% (1.4-6.4) | 1.5% (0.5-4.3) | |
| Tyva Republic | 210 | 8 | 1 | 7 | 0 | 0 |
| | | 3.8% (1.9-7.3) | 0.5% (0.1-2.6) | 3.3% (1.6-6.7) | | |
| Chita region | 150 | 3 | 3 | 0 | 0 | 0 |
| | | 2.0% (0.7-5.7) | 2.0% (0.7-5.7) | | | |
| Khakassia Republic | 150 | 3 | 0 | 3 | 0 | 0 |
| | | 2.0% (0.7-5.7) | | 2.0% (0.7-5.7) | | |
| PrimorskyKrai | 368 | 4 | 3 | 0 | 0 | 1 |
| | | 1.1% (0.4-2.8) | 0.8% (0.3-2.3) | | | 0.3% (0.1-1.5) |
| **Total:** | 1078 | 27 | 7 | 16 | 3 | 1 |
| | | 2.5% (1.7-3.6) | 0.6% (0.3-1.3) | 1.5% (0.9-2.4) | 0.3% (0.1-0.8) | 0.1% (0.02-0.5) |

MKWV isolates ranges from 83% to 91%, consistent with their classification within the broader MKWV species complex but highlighting distinct genetic divergence between the lineages. These findings support the notion that Gomselga virus represents a phylogenetically coherent subgroup within the MKWV species, circulating widely across Russia and maintaining significant genetic homogeneity despite geographic dispersion.

The MJPV Primorye-29 isolate (GenBank ID: PP476844) identified in this study clustered robustly within a distinct clade together with Chinese MJPV isolates in phylogenetic analysis. The nucleotide sequence identity in the RdRp-encoding fragment between Primorye-29 and these Chinese isolates ranged from 91% to 99%, indicating close genetic relatedness.

The OTPV isolates identified in this study clustered phylogenetically with previously known isolates from Russia and China, forming a well-supported monophyletic group. They exhibited a high degree of nucleotide sequence identity (96–100%) in the RdRp-encoding fragment, indicating strong genetic conservation within this viral lineage. This close relationship suggests ongoing circulation of a genetically stable OTPV clade across the region and underscores its established presence in natural foci of Russia.

Given the high genetic homogeneity observed among the OTPV isolates and their close clustering with previously characterized strains, further full-genome sequencing was deemed less critical for understanding their evolutionary dynamics. Similarly, due to the limited number of MJPV-positive ticks and the fragmented nature of the obtained sequences, complete genome assembly was not feasible. In contrast, MKWV exhibited notable genetic diversity across its genomic segments, particularly among isolates from different regions, and showed evidence of potential reassortment and distinct phylogeographic patterns. Therefore, priority was given to obtaining full-length genomic sequences of MKWV to enable comprehensive phylogenetic, structural, and functional analyses, which are essential for elucidating the molecular epidemiology and evolutionary history of this emerging phlebovirus in Russia and beyond.

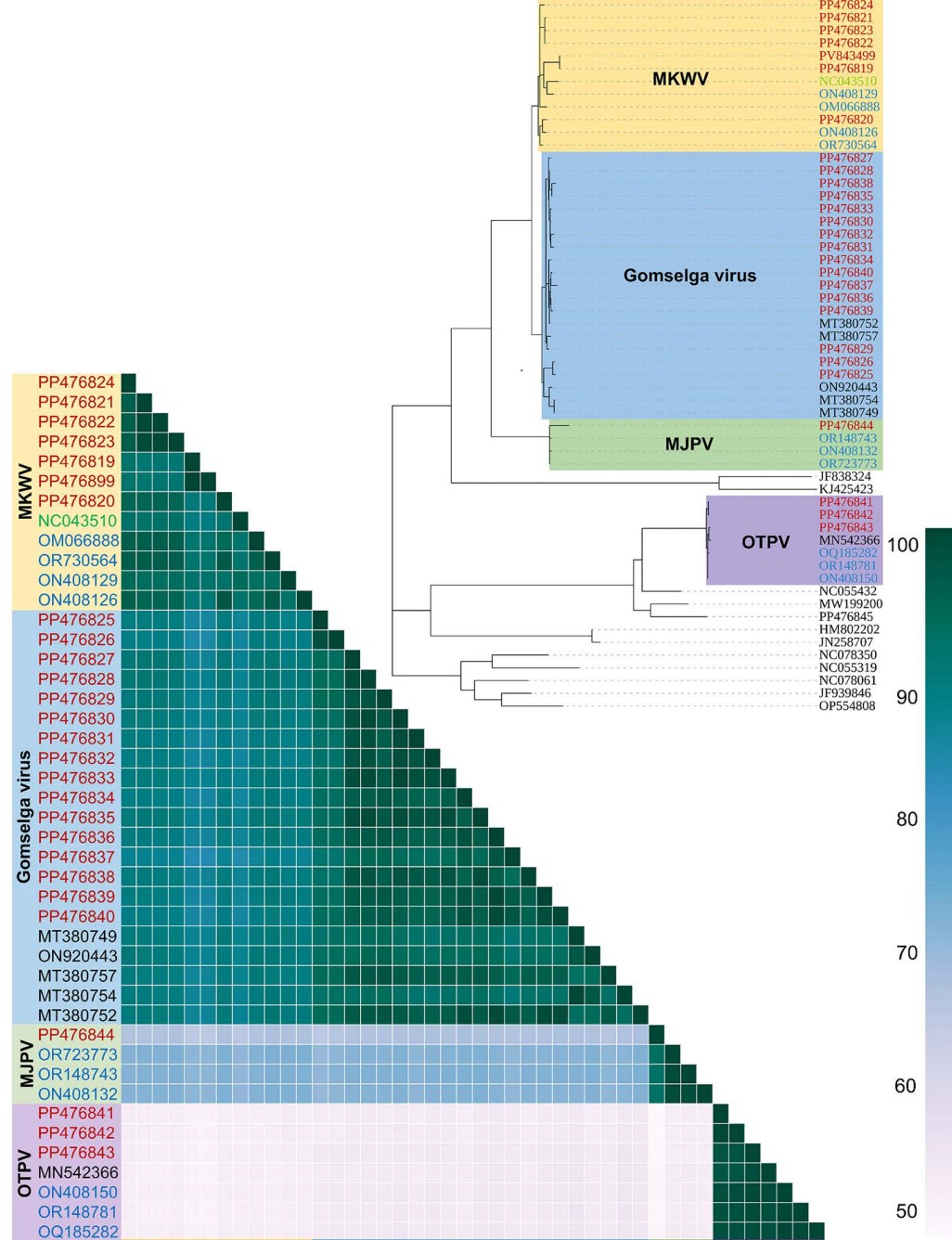

**Fig 1. Maximum likelihood molecular phylogenetic analysis and pairwise sequence identity matrix of the partial tick phlebovirus L segment sequences.** The analysis involved 56 nucleotide sequences. There were a total of 401 positions in the final dataset. All positions containing gaps and missing data were eliminated. GenBank accession numbers for phleboviral sequences identified in this study are indicated in red, those for other previously reported sequences from Russia are shown in black, Chinese isolates are labeled in blue, and the isolate from Japan is highlighted in green.

## Molecular characterization of identified Mukawa virus complete coding sequences

To determine the full-length MKWV protein-coding sequences of each genomic segment, a primer panel was designed and optimized. Primer selection was based on a multiple sequence alignment of all available MKWV complete protein-coding sequences of retrieved from GenBank, ensuring broad coverage across the genetic diversity of the virus. The primer panel was empirically optimized through a series of amplification tests, with the optimal annealing temperature determined to be 56 °C, using a touchdown PCR protocol with a decrement of 0.1 °C per cycle to enhance specificity and yield. This optimized approach enabled efficient amplification of the MKWV S, M, and L segments, facilitating downstream sequencing and comprehensive genomic characterization of MKWV variants circulating in southern Siberia.

## Mukawa virus genome of Russian isolates

The complete coding sequences of seven Russian MKWV isolates obtained in this study were placed in GenBank under accession numbers: PP525061-PP525081. The observed Russian MKWV isolates exhibit the characteristic tripartite, segmented genome organization typical of phleboviruses.

We applied a structure-based approach leveraging AI-driven predictions for the MKWV genome annotation. The generated MKWV protein structure models exhibited high confidence metrics (pTM and pLDDT), enabling reliable and precise genome annotation (Fig 2).

The S segment utilizes an ambisense coding strategy to encode both the nucleoprotein (N, 247 a.a., pTM = 0.89, pLDDT = 0.95) and a nonstructural (NS) protein (NS, 340 a.a., pTM = 0.64, pLDDT = 0.66) (Fig 2, S3 Table). The high pLDDT and pTM values for MKWV N indicate a well-predicted, structurally confident model, consistent with its conserved role in RNA encapsidation.

The M segment encodes a glycoprotein precursor (GPC, 966 a.a., pTM = 0.59, pLDDT = 0.79) that is post-translationally cleaved into the two viral surface glycoproteins: G1 (522 a.a., pTM = 0.53, pLDDT = 0.80) and G2. Domain-specific structural predictions were performed for MKWV G2, with separate modeling of the G2 fusion domain (312 a.a., pTM = 0.81, pLDDT = 0.90) and the G2 C-terminal transmembrane domain (86 a.a., pTM = 0.77, pLDDT = 0.90) (Fig 2, S3 Table).The high pLDDT and pTM scores across both glycoprotein domains indicate strong structural confidence, supporting their functional roles in membrane fusion and virion assembly, and reinforcing the conserved molecular architecture of phlebovirus glycoproteins. Key structural features of the MKWV G2 fusion domain are consistent with the canonical class II viral fusion fold. The domain contains a core rich in β-sheet structure. In our G2 sequence, the fusion loop is predicted to reside at the tip of domain 2, spanning approximately residues 670–684. The amino acid sequence of this putative fusion loop is QCGGAGCGCFNIHAS (S1 and S2 Fig).

The L segment encodes the large RNA-dependent RNA polymerase complex (RdRp complex, 2098 a.a., pTM = 0.84, pLDDT = 0.79). Within this multifunctional protein, structurally conserved domains were identified, including the N-terminal endonuclease domain (182 a.a.; pTM = 0.89, pLDDT = 0.93) and the central RdRp catalytic domain (697 a.a.; pTM = 0.94, pLDDT = 0.96). In addition, three regions with no currently assigned functional role were predicted to form stable, well-defined structural domains with high confidence ($0.51 < pTM < 0.89$, $0.77 < pLDDT < 0.89$), suggesting potential roles in viral replication or host interaction that remain to be experimentally characterized (Fig 2, S3 Table). The high structural reliability of these domains supports their biological relevance and warrants further investigation into their functional significance within the MKWV replication machinery. This genomic architecture is consistent with that of other members of the *Phlebovirus* genus and confirms the functional integrity of the newly sequenced MKWV genomes.

## MKWV nucleoprotein structure analysis

The MKWV nucleoprotein was selected for detailed structural analysis due to its high evolutionary conservation and critical role in ribonucleoprotein complex assembly.

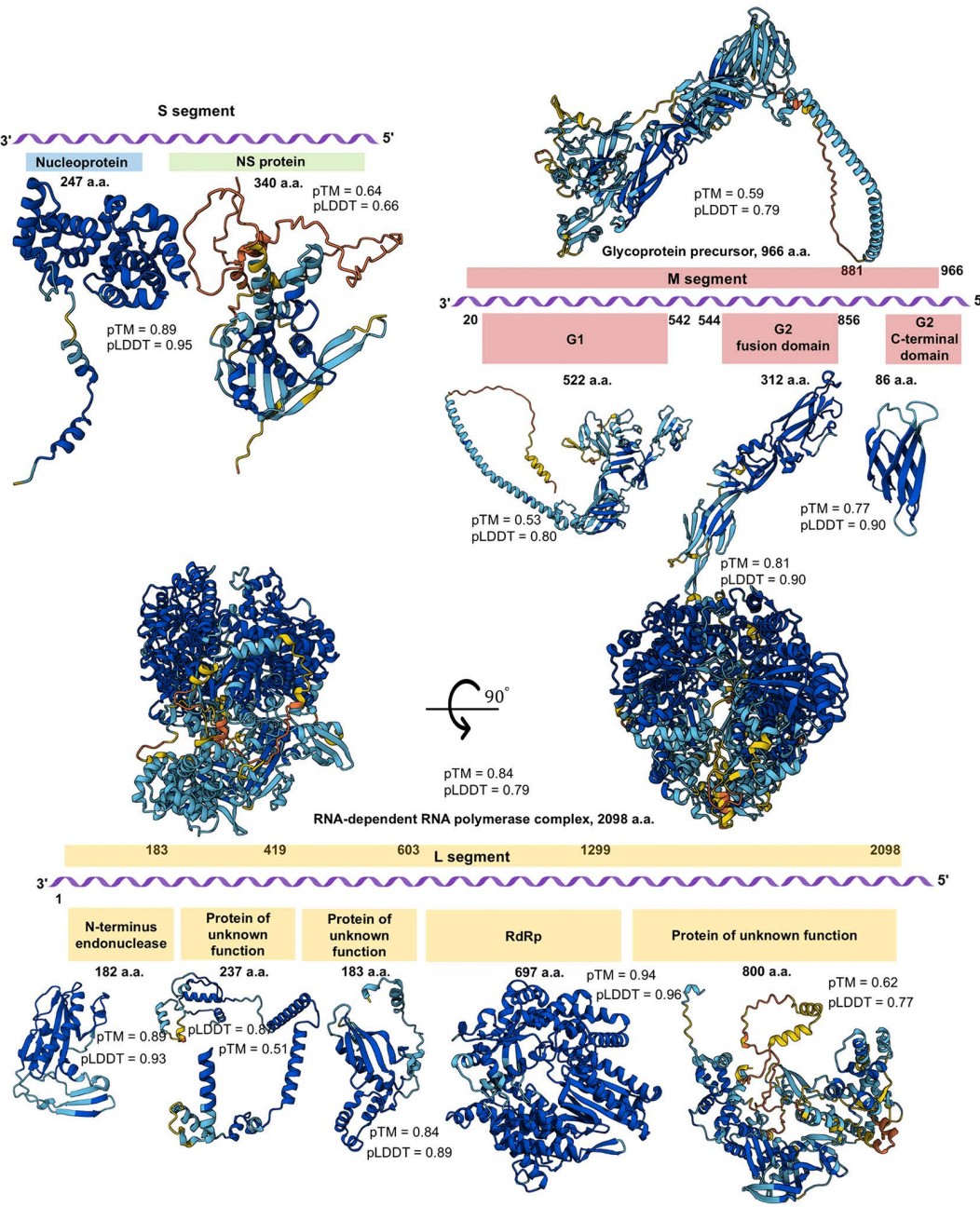

**Fig 2. Schematic genome organization of Russian MKWV isolates.** The MKWV genomic ssRNA(-) is depicted in purple. Putative proteins and functional domains encoded by the Russian MKWV isolates are shown as follows: nucleoprotein (blue) and NS protein (green) on the S segment, glycoprotein precursor (red) on the M segment and RNA-dependent RNA polymerase complex (yellow) on the L segment. Predicted spatial structures of the MKWV encoded proteins are illustrated with color-coding based on pLDDT scores (per-residue confidence values from AlphaFold predictions), ranging from blue (pLDDT>90, high confidence) to red (pLDDT<50, low confidence), indicating the reliability of the structural models across different domains. For each MKWV protein structure model, numerical values of both the pLDDT score and the pTM score are provided, reflecting local and global confidence levels of the structural predictions, respectively. All structural models were generated using the amino acid sequences of the MKWV Tuva-46 isolate as the reference template.

The predicted structure of the MKWV nucleoprotein can be divided into three distinct domains. Residues 1–32 form a flexible N-terminal region containing two α-helical segments that extend outward from the globular core of the protein. The globular core itself consists of two structural domains: one domain is composed of six α-helices spanning residues 37–92, 114–125, and 213–222, while the second domain comprises six α-helices located at residues 106–112 and 132–222 (Fig 3). The MKWV nucleoprotein core domain is structurally highly similar to the previously reported crystal structures of the Rift Valley fever virus nucleoprotein (*Phlebovirus riftense*, PDB ID: 4H5O, TM-score = 0.901), Toscana virus nucleoprotein (*Phlebovirus toscanaense*, PDB ID: 4CSF, TM-score = 0.908), and Buenaventura virus nucleoprotein (*Phlebovirus buenaventuraense*, PDB ID: 4J4W, TM-score = 0.900), despite limited sequence identity (~40%) (S3 Table).

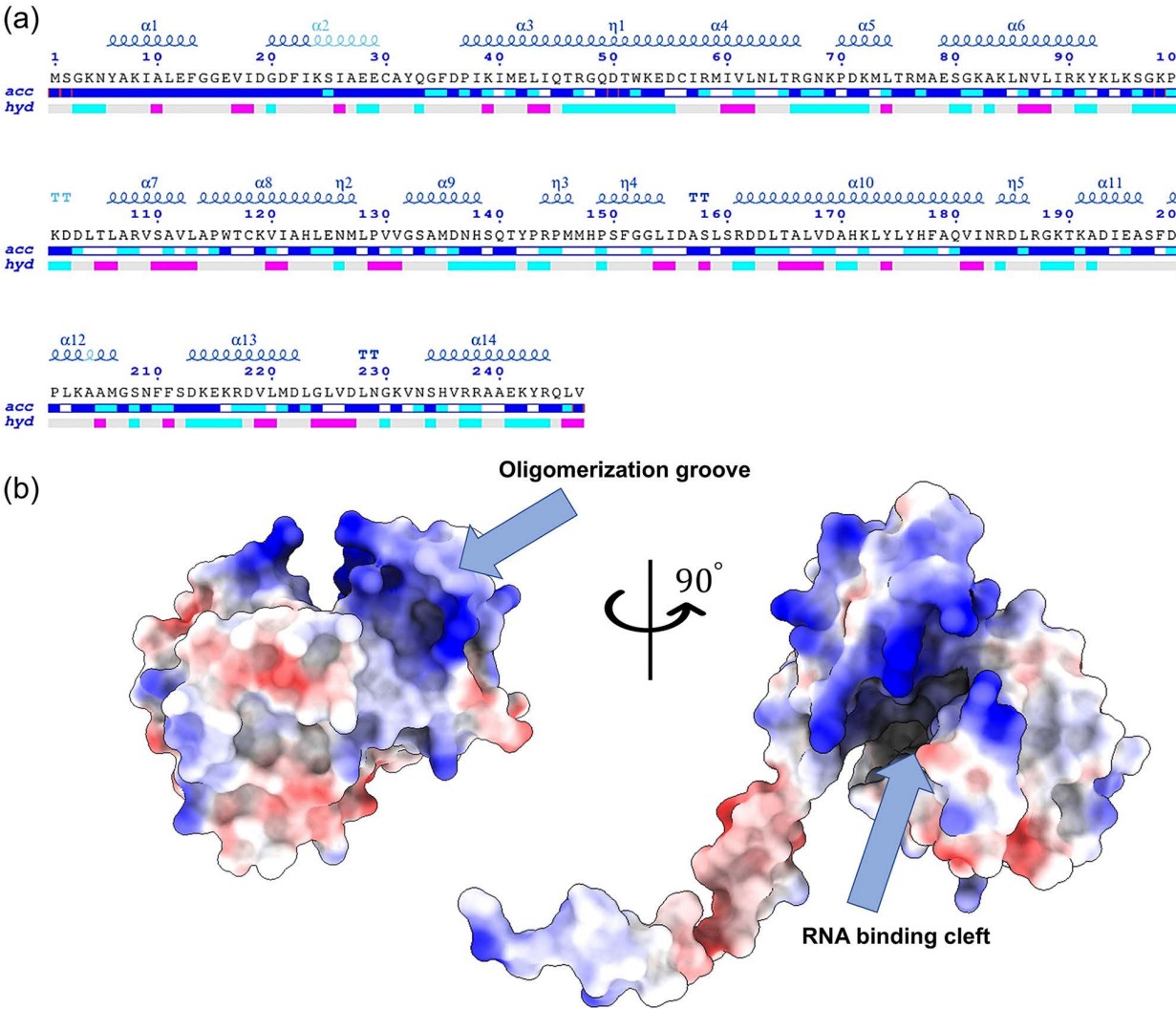

**Fig 3. Structural characterization of the MKWV nucleoprotein. (a)** Sequence of the MKWV nucleoprotein with secondary structure elements. The primary sequence of the MKWV nucleoprotein is depicted alongside its predicted secondary structure, highlighting α-helices as determined by AlphaFold-based structural modeling. The relative accessibility (labelled "acc") calculated by DSSP for each residue is shown with a colored bar below the sequences block: white is buried, cyan is intermediate, blue is accessible. The hydropathy (labelled "hyd") calculated from the query sequence using the Kyte&Doolittle algorithm is shown by a second colored bar below the accessibility: pink is hydrophobic, grey is intermediate and cyan is hydrophilic. **(b)** Putative model of the MKWV nucleoprotein structure in electrostatic surface potential representation. The positive surface potential is colored blue, and the negative surface potential is colored red.

The root-mean-square deviation (RMSD) between the backbone atoms of the Rift Valley fever virus and MKWV nucleoprotein structures is approximately 2.0 Å, indicating a nearly identical fold (S3 Fig). The highly basic MKWV nucleoprotein RNA-binding cleft and oligomerization groove, enriched in positively charged residues, is clearly visible and structurally conserved relative to other phleboviral nucleoproteins, suggesting a critical role in encapsidation of the viral genomic RNA (Fig 3). This structural insight supports the functional annotation of the MKWV nucleoprotein as a key component of the viral ribonucleoprotein complex.

## Phylogenetic and homology analysis of MKWV genomic segments

Phylogenetic analysis based on all three MKWV genomic segments (L, M, and S) revealed varying levels of genetic homogeneity among the MKWV isolates. Three isolates from Chita region (GenBank ID: PP525076-PP525078, PP525069-PP525071, PP525062-PP525064) exhibited high genetic consistency, consistently forming a distinct and well-supported subclade across all three segments, suggesting local circulation of a closely related lineage in this region (Fig 4 (a)).

In contrast, MKWV Tuva-46 isolate (GenBank ID: PP525075, PP525068, and PP525061) displayed a more complex phylogenetic pattern. While it clustered with a single MKWV Primorye-50 isolate (GenBank ID: PP525080) from Primorsky Krai in the S segment tree, it grouped robustly with prototype Chinese MKWV strains: XQ1 (GenBank ID: OR730569) and YKS1 (GenBank ID: OR730568) in the M and L segment analyses, indicating closer genetic relatedness to these reference strains (Fig 4, S4 Table).

The highest degree of genetic diversity was observed among MKWV variants circulating in Primorsky Krai. These isolates showed incongruent clustering patterns across the three genomic segments, indicative of potential genetic reassortment or regional diversification. In the M segment analysis, all Primorsky MKWV isolates (GenBank ID: PP525072-PP525074) formed a unified subclade, closely grouping with the prototype MKWV NE-YC4 strain (GenBank

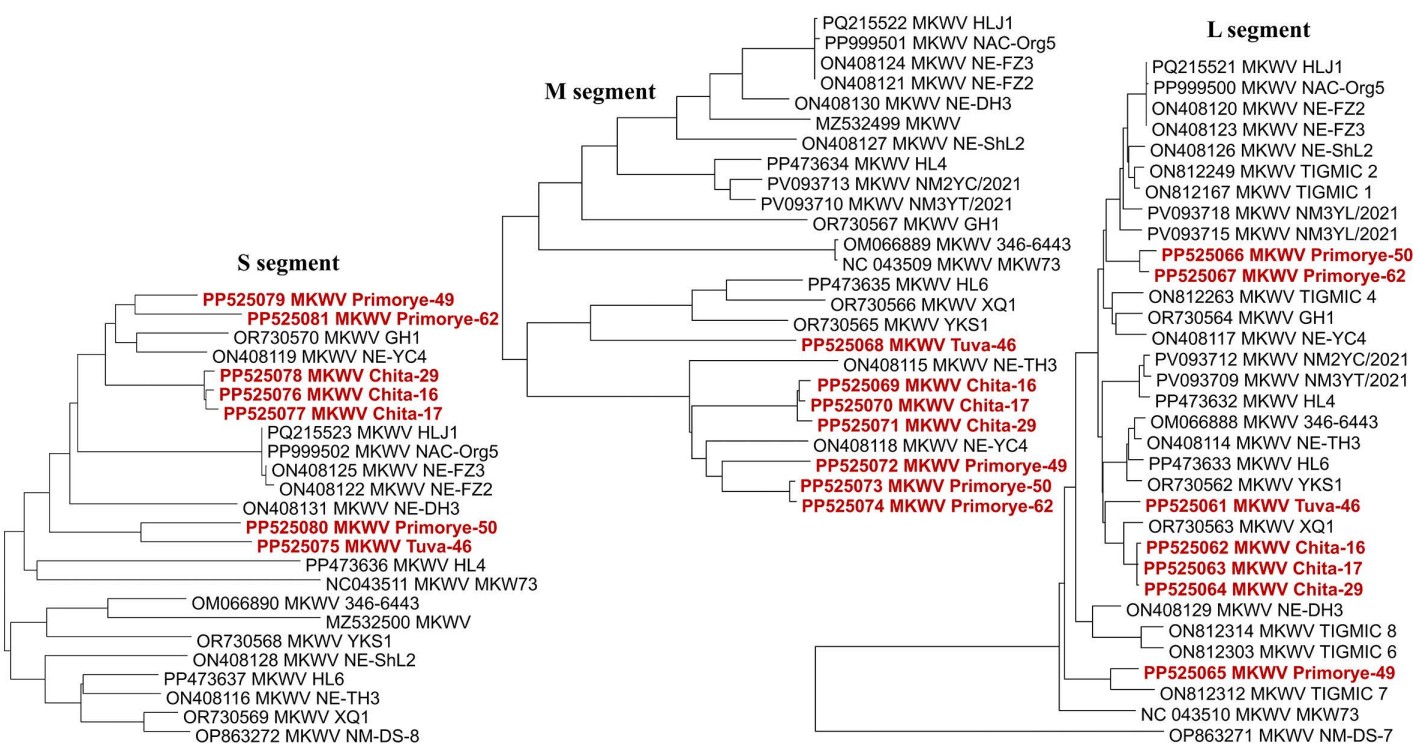

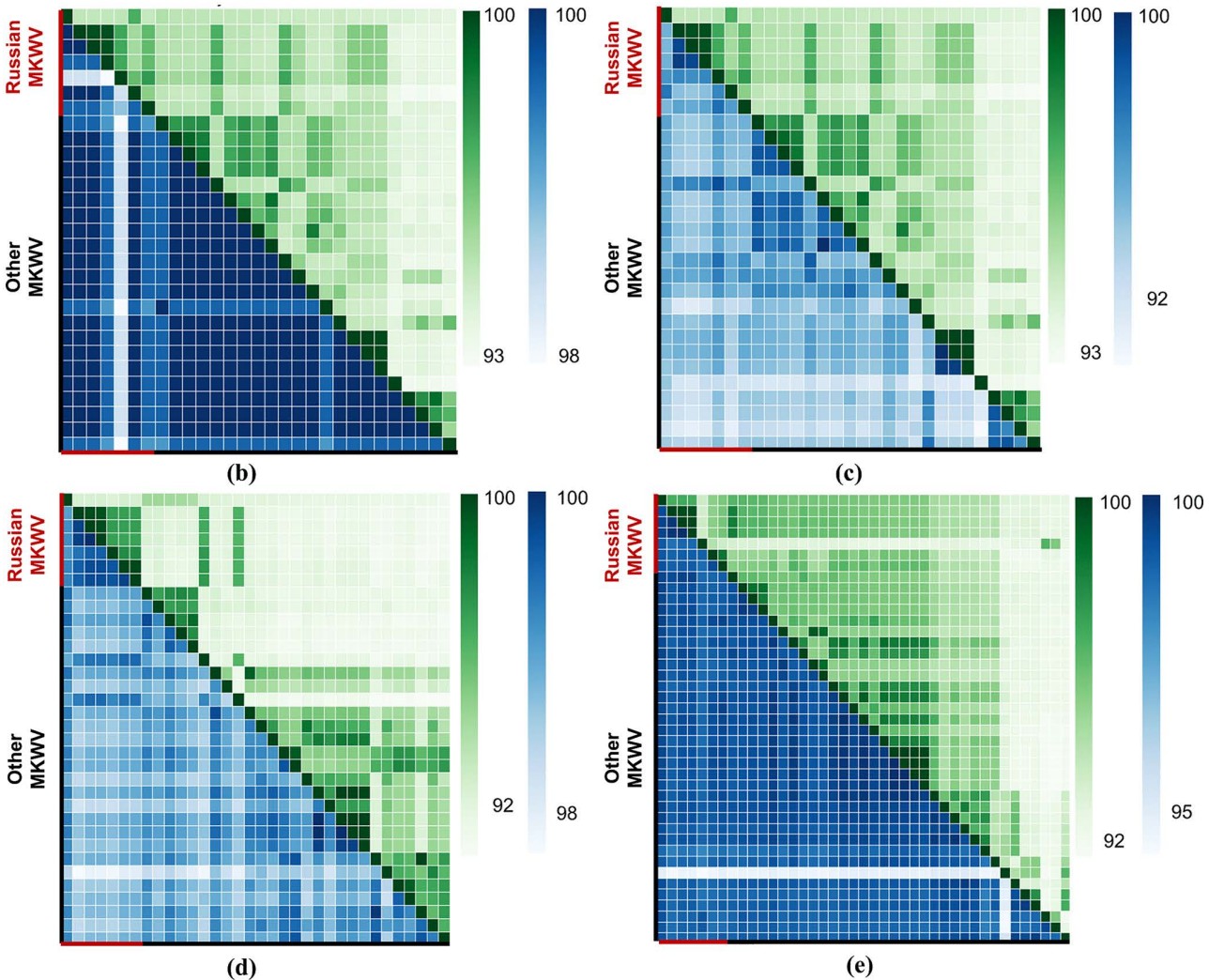

**Fig 4. Maximum likelihood molecular phylogenetic analysis of the full-length MKWV protein-coding sequences (a) and heatmaps based on MKWV nucleotide (from bright green to light green) and amino acid (from bright blue to light blue) sequence genetic distances (b-e): (b)** – nucleoprotein sequences of S segment, **(c)** – NS protein sequences of S segment, **(d)** – GPC sequences of M segment, and **(e)** – RdRp complex sequences of L segment.The phylogenetic analysis involved 24 nucleotide sequences for MKWV S segment (1834 positions in the final dataset), 25 nucleotide sequences for MKWV M segment (2898 positions in the final dataset), and 33 nucleotide sequences for MKWV L segment (5970 positions in the final dataset). All positions containing gaps and missing data were eliminated. MKWV sequences of Russian isolates identified in this study are highlighted in red.

ID: ON408119) from Heilongjiang, China, as well as other Russian MKWV variants from Chita region, suggesting shared ancestry in the glycoprotein-coding region. However, in the S segment tree, two of the Primorsky MKWV isolates (Primorye-49 and Primorye-62) formed a distinct subclade closely related to Chinese MKWV GH1strain (GenBank ID: OR730570) and Chinese MKWV NE-YC4 strain (GenBank ID: ON408119), while MKWV Primorye-50 isolate clustered instead with the MKWV Tuva-46 isolate. A similar discordant pattern was observed in the L segment phylogeny: two iso-lates formed a cohesive group, whereas MKWV Primorye-49 isolate was phylogenetically divergent and grouped with the prototype strain MKWV TIGMIC_7 isolate (GenBank ID: ON812312) from China (Fig 4, S4 Table).

These discrepancies in topological placement across segments suggest that MKWV populations in Primorsky Krai may be undergoing active evolution, potentially involving reassortment events or localized adaptive divergence, warranting further investigation into the molecular mechanisms driving phlebovirus diversity in this region.

In addition, we observed that MKWV Primorye-49 isolate harbors four amino acid substitutions in the nucleoprotein – the most conserved protein among phleboviruses (S4 Table). However, structural modeling indicates that these substitutions are located on surface and do not alter the overall nucleoprotein tertiary structure (Fig 5, S4 Fig). Given the preservation of the core structural architecture, these mutations are unlikely to impair essential functions, such as RNA encapsidation or ribonucleoprotein complex formation. Nevertheless, their presence in a genetically divergent isolate highlights regions of potential antigenic or protein-interaction variability that may contribute to viral fitness or immune evasion, meriting further functional characterization.

## Discussion

The detection of MJPV in Primorsky Krai and OTPV in Irkutsk region confirms their circulation within the territory of Russia. The newly identified MJPV isolates showed 91–99% nucleotide sequence identity with Chinese isolates. The OTPV isolates exhibited even higher similarity, ranging from 96–100% identity with Chinese isolates. This close genetic relationship likely reflects the geographic proximity of these regions to China, which facilitates the cross-border movement of ticks

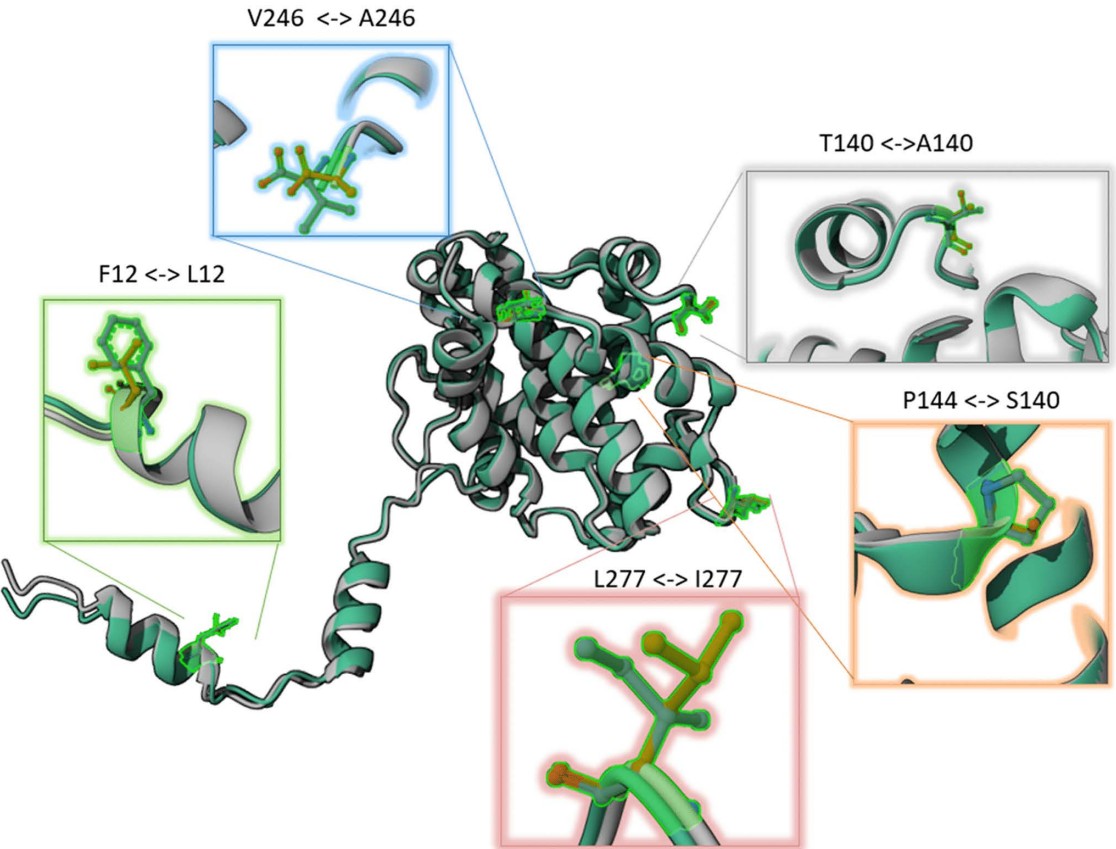

**Fig 5. Superposition of the nucleoprotein tertiary structure models from MKWV Tuva-46 (gray) and Primorye-49 (green) isolates.** Amino acid substitutions specific to the MKWV Primorye-49 isolate are highlighted in colored boxes.

and reservoir hosts, such as migratory birds. However, it is important to note that these similarities were primarily based on partial genome fragments, which may not fully capture the evolutionary signal. These findings confirm viral presence in Russia that had previously been only hypothesized [26]. The relatively low detection rates of MJPV and OTPV observed in this study may have several explanations. First, these viruses may have been recently introduced into the surveyed regions. Second, they may have a lower prevalence within local tick populations compared to other phleboviruses. Third, MJPV and OTPV may show strict ecological association with specific biotopes that were not comprehensively sampled in the present work.

Our findings extend the known geographic range of MKWV-related variants to several Russian regions and provide evidence that previously described Gomselga virus isolates belong within the broader MKWV species complex. This taxonomic re-evaluation, supported by phylogenetic analysis and high sequence identity, suggests that these viruses represent regional lineages of a single, genetically diverse species (Fig 1).

We observed incongruence in phylogenetic placement of certain MKWV isolates when analyzed per segment, especially among samples from Primorsky Krai. This topological discordance, combined with the identification of unique amino acid substitutions in the otherwise highly conserved nucleoprotein of Primorye-49 isolate, points to possibly reassortment events within local tick populations. Reassortment has been documented in other phleboviruses and may accelerate adaptation to new vectors or hosts [43,44]. Although no structural alterations were detected in the core domains of the nucleoprotein, the presence of surface-exposed substitutions warrants further investigation into their potential impact on antigenicity or host immune recognition.

The successful application of AI-driven structural modeling significantly enhanced genome annotation accuracy, allowing confident prediction of functional domains in the N, GPC, and RdRp. These models serve as a foundation for future structure-function studies and may inform antiviral design targeting conserved viral components.

The structural modeling of the MKWV nucleoprotein deserves particular attention. The high degree of structural homology with the crystal structures of nucleoproteins from other phleboviruses (Rift Valley fever virus, Toscana virus, etc.), demonstrated in this study despite limited primary sequence homology, confirms the evolutionary conservation of the tertiary structure of this protein (S3 Table, S3 Fig). The well-defined RNA-binding cleft, enriched with positively charged amino acid residues, is critical for interaction with the viral genome. The preservation of the overall tertiary structure even in the phylogenetically divergent isolate Primorye-49, which carries four amino acid substitutions in the nucleoprotein, indicates strict evolutionary constraints aimed at preserving its function. This represents a classic example of structural evolution, where changes in the sequence compensate for each other to maintain the integrity of the final spatial structure. Nevertheless, the localization of these substitutions on the protein surface suggests their potential involvement in modulating protein-protein interactions or immune recognition, which warrants experimental verification.

An important aspect is the co-circulation of multiple tick-borne pathogens in natural foci. In endemic regions of Russia, the proportion of ticks infected with two or more pathogens can reach 10–30% or higher [45,46]. Coinfection poses significant challenges for clinical diagnosis and often results in atypical or more severe disease presentations. Mixed infections have been reported to cause more pronounced fever and systemic toxicity compared to mono-infections caused by individual agents. Moreover, the simultaneous presence of different pathogens in a tick or vertebrate host may modulate the expression of virulence factors, potentially altering pathogenicity and transmission dynamics [47,48]. In this work, all tick samples were additionally screened via PCR for a panel of clinically relevant tick-borne agents, including TBEV, Kemerovo virus (*Orbivirus magninsulae*), YEZV, ALSV, *Borrelia burgdorferi*, *Borrelia miyamotoi*, *Anaplasma phagocytophilum*, and *Rickettsia spp*. Notably, coinfection was detected in several specimens. MKWV Chita-17 isolate was found together with ALSV (GenBank ID: PP623704, PP623719, PP623749) in the same tick, while MKWV Primorye-49 and Primorye-50 isolates were co-detected with *Borrelia miyamotoi* (GenBank ID: PV652158 and PV652159, accordingly). Furthermore, dual infection with MKWV and *Rickettsia spp*. was observed in ticks harboring MKWV Tuva-46 and Chita-16 isolates, underscoring the broad microbial complexity of tick-borne pathogen communities. These findings underscore the

complexity of tick microbiomes in nature and emphasize the need for multiplex diagnostic approaches in clinical settings, where overlapping symptomatology could mask underlying coinfections. They also highlight the importance of integrated surveillance programs capable of detecting both known and emerging pathogens simultaneously.

This work demonstrates the value of integrating high-throughput sequencing, phylogenetics, and structural bioinformatics into arbovirus surveillance programs. Expanding such efforts is essential for early detection of novel viruses and viral strains, assessment of zoonotic potential, and preparedness against future outbreaks.

## Conclusion

In this study, we extended the known geographic range of recently discovered MKWV, Gomselga virus, OTPV, and MJPV to several regions of Asian Russia. Furthermore, we determined complete coding sequences of the identified MKWV isolates, annotated the genome using protein structure predictions, and analyzed the nucleoprotein tertiary structure of Russian MKWV isolates. These findings contribute to a better understanding of the genetic diversity and phylogeography of these pathogens within Russia and have direct practical implications for the epidemiological surveillance system. Up-to-date information on the circulation patterns and genetic variability of phleboviruses with potential zoonotic risk for humans and animals supports the optimization and adjustment of preventive and epidemic control measures. The detection of phlebovirus circulation in *Ixodes persulcatus* ticks across the vast territory of Eastern Siberia and Far East underscores the necessity of continued monitoring of phlebovirus distribution in natural foci, particularly in regions bordering China. Such surveillance is essential for early detection of emerging viral threats and assessment of their public health significance.

## Supporting information

**S1 Table. Oligonucleotide primers used for targeted library enrichment on NGS.**
(PDF)

**S2 Table. Sequences using for analysis in this study.**
(PDF)

**S3 Table. Comparison of MKWV putative protein tertiary structures with those of homologous proteins from closely related viruses.**
(PDF)

**S1 Fig. The putative fusion loop of Gc MKWV Tuva-46 isolate (hydrogen bonds are shown in blue).**
(TIF)

**S2 Fig. Comparison of the Gc protein from Mukawa virus (MKWV, isolate Tuva-46, colored violet) with the crystal structure of glycoprotein C from Rift Valley fever virus (PDB: 4HJ1, shown in blue).**
(TIF)

**S3 Fig. Superposition of the MKWV Tuva-46 nucleoprotein tertiary structure model (pLDDT color) and crystal structure of Rift Valley fever virus nucleoprotein (PDB ID: 4H5O, gray).**
(TIF)

**S4 Table. Identity level of nucleotide and amino acid sequences (%) of identified Russian MKWV isolates with a MKWV prototype isolates.**
(XLSX)

**S4 Fig. Superposition of the MKWV nucleoprotein tertiary structure models from Russian isolates.**
(TIF)

## Acknowledgments

The authors are grateful to the colleagues who conducted tick collection in the surveyed territories of the Russian Federation: staff of the Hygienic and Epidemiological Centers of Irkutsk Region (Chief Physician I.V. Bezgodov), Primorsk Region (Acting Chief Physician E.V. Pyatyrova), the Altai Republic (Chief Physician G.S. Arkhipov), Tyumen Region (Chief Physician A.Ya. Folmer), Kemerovo Region (Chief Physician A.V. Bachina), Krasnoyarsk Region (Chief Physician D.A. Khodov), and the Tyva Republic (zoologist S.S. Saryglar); as well as staff of the Primorsk Anti-Plague Station (Director N.S. Gordeyko), Khabarovsk Anti-Plague Station (Director A.G. Kovalsky), and Chita Anti-Plague Station (Director A.B. Moshkin).

## Author contributions

**Conceptualization:** Mikhail Y. Kartashov, Anastasia V. Gladysheva.

**Data curation:** Mikhail Y. Kartashov.

**Formal analysis:** Mikhail Y. Kartashov, Anastasia V. Gladysheva.

**Funding acquisition:** Alexander P. Agafonov.

**Investigation:** Mikhail Y. Kartashov, Valentina Y. Kurushina, Alexey O. Yanshin, Tatyana V. Tregubchak, Alina S. Zheleznova, Kirill A. Svirin, Vladimir A. Ternovoi, Alexander P. Agafonov, Anastasia V. Gladysheva.

**Methodology:** Mikhail Y. Kartashov, Valentina Y. Kurushina, Alexey O. Yanshin, Alina S. Zheleznova, Kirill A. Svirin.

**Project administration:** Vladimir A. Ternovoi, Alexander P. Agafonov.

**Resources:** Mikhail Y. Kartashov, Vladimir A. Ternovoi, Alexander P. Agafonov.

**Supervision:** Mikhail Y. Kartashov, Anastasia V. Gladysheva.

**Visualization:** Mikhail Y. Kartashov, Alexey O. Yanshin, Anastasia V. Gladysheva.

**Writing – original draft:** Mikhail Y. Kartashov, Anastasia V. Gladysheva.

**Writing – review & editing:** Mikhail Y. Kartashov, Valentina Y. Kurushina, Alexey O. Yanshin, Tatyana V. Tregubchak, Alina S. Zheleznova, Kirill A. Svirin, Vladimir A. Ternovoi, Alexander P. Agafonov, Anastasia V. Gladysheva.

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
