## [Decision Letter · Decision Letter 0]

1 Apr 2026

PONE-D-26-10722Expanding diversity of tick-borne phleboviruses (Phlebovirus mukawaense, Mudanjiang phlebovirus, Gomselga Virus, and Onega tick phlebovirus) in RussiaPLOS One

Dear Dr. Gladysheva,

Thank you for submitting your manuscript to PLOS ONE. After careful consideration, we feel that it has merit but does not fully meet PLOS ONE’s publication criteria as it currently stands. Therefore, we invite you to submit a revised version of the manuscript that addresses the points raised during the review process.

A letter that responds to each point raised by the academic editor and reviewer(s). You should upload this letter as a separate file labeled ’Response to Reviewers’.A marked-up copy of your manuscript that highlights changes made to the original version. You should upload this as a separate file labeled ’Revised Manuscript with Track Changes’.An unmarked version of your revised paper without tracked changes. You should upload this as a separate file labeled ’Manuscript’.

We look forward to receiving your revised manuscript.

Kind regards,

Humberto Julio Debat

Academic Editor

PLOS One

**Journal Requirements:**

1. Please ensure that your manuscript meets PLOS ONE’s style requirements, including those for file naming. The PLOS ONE style templates can be found at

“The work was supported by the Ministry of Science and Higher Education of the Russian Federation (The Federal scientific-technical program for genetic technologies development for 2019-2030, Agreement № 075-15-2025-526).”

4. Please note that your Data Availability Statement is currently missing a direct link to access each database. If your manuscript is accepted for publication, you will be asked to provide these details on a very short timeline. We therefore suggest that you provide this information now, though we will not hold up the peer review process if you are unable.

5. We note that Figure 1 in your submission contain map images which may be copyrighted. All PLOS content is published under the Creative Commons Attribution License (CC BY 4.0), which means that the manuscript, images, and Supporting Information files will be freely available online, and any third party is permitted to access, download, copy, distribute, and use these materials in any way, even commercially, with proper attribution. For these reasons, we cannot publish previously copyrighted maps or satellite images created using proprietary data, such as Google software (Google Maps, Street View, and Earth). For more information, see our copyright guidelines: http://journals.plos.org/plosone/s/licenses-and-copyright.

6. Please upload a new copy of Figures 2, 3, 4, 5, and 6 as the detail is not clear. Please follow the link for more information:  https://journals.plos.org/plosone/s/figures

7. We note that there is identifying data in the Supporting Information file < S1. Table.pdf >. Due to the inclusion of these potentially identifying data, we have removed this file from your file inventory. Prior to sharing human research participant data, authors should consult with an ethics committee to ensure data are shared in accordance with participant consent and all applicable local laws.

-Location data

Please remove or anonymize all personal information (location data), ensure that the data shared are in accordance with participant consent, and re-upload a fully anonymized data set. Please note that spreadsheet columns with personal information must be removed and not hidden as all hidden columns will appear in the published file.

Reviewers’ comments:

Reviewer’s Responses to Questions

**Comments to the Author**

1. Is the manuscript technically sound, and do the data support the conclusions?

Reviewer #1: Yes

Reviewer #2: Yes

Reviewer #3: Yes

2. Has the statistical analysis been performed appropriately and rigorously? 

Reviewer #1: Yes

Reviewer #2: N/A

Reviewer #3: Yes

3. Have the authors made all data underlying the findings in their manuscript fully available?

Reviewer #1: Yes

Reviewer #2: Yes

Reviewer #3: Yes

4. Is the manuscript presented in an intelligible fashion and written in standard English?

Reviewer #1: No

Reviewer #2: Yes

Reviewer #3: Yes

5. Review Comments to the Author

Reviewer #1: The manuscript is of interest because it provides new information about the circulation of phleboviruses in Russia.

Below I present my comments, eliminating which the authors can improve their manuscript.

If you describe the segments from which translationally processed proteins are synthesized, you should indicate, including in the figures, the sites of cleavage or other modifications.

When describing the M segment, to understand the biology of the virus, it is more appropriate to look at the structure of the Gc and Gn proteins separately, rather than the precursor. In the table, you also describe the Gc fusion domain, but you don’t provide its structure anywhere. What are its features, and what about the fusion loop? This should be examined in more detail.

Figure 3 requires a clearer description of the structures shown. The lower part of the figure is unclear. In the text you describes RdRp and regions 182, 657, and "in addition," three regions with no currently assigned functional role. However, the figure also includes regions with amino acid length 293 and 800. What is this? These domains are not listed in Table S4. The figure has no description at all for the regions highlighted in gray.

Further in the text is a description of the nucleoprotein structure analysis. Why was this protein specifically given such a detailed analysis, while others, such as Gc, were not? In Figure 4, it’s also worth separating the top and bottom halves of the figure, labeling them A and B, and providing a separate description for each part.

Reviewer #2: This study provides valuable information regarding tick-associated phleboviruses, specifically addressing geographical variability, prevalence rates, and relatedness to other zoonotic potential lineages. The data and supporting information are well-presented, and the scientific findings warrant publication.

The Introduction is exceptionally well-written, offering a clear historical context for phlebovirus in Asia/Russia and surrounding regions.

Below are minor comments need the author’s attention:

Methodology:

Lines 128-129: Please provide more detail on the tick sample homogenization process. Given the hard exoskeleton, a physical force step is necessary to initially break open the tick before further processing.

Lines 132-138: RNA stability is more fragile than DNA. The successful PCR amplification of DNA target does not guarantee the integrity of the original RNA template. This point requires clarification.

Results/Figures/Tables:

Table 1: The percentage of infection is missing for the Mukawa virus under the Chita region.

Figures: The quality of the pictures need improvement, particularly the clarity of the dots representing each detected virus (Fig 1).

General:

Please thoroughly check the entire manuscript for typos and correct all mistakes. Examples of lines needing attention include 316 and 371.

Reviewer #3: The manuscript characterizes the prevalence of four tick-borne viruses in Ixodes persulcatus ticks in Asian Russia. The research is well-designed and nicely-crafted. Some minor comments were raised in the uploaded file, and the authors are requested to go through it and act accordingly.

6. PLOS authors have the option to publish the peer review history of their article (what does this mean?). If published, this will include your full peer review and any attached files.

Reviewer #1: No

Reviewer #2: No

Reviewer #3: No

---

## [Author Response · Author response to Decision Letter 1]

29 Apr 2026

Thank you for reviewing our manuscript (PONE-D-26-10722) entitled “Expanding diversity of tick-borne phleboviruses (Phlebovirus mukawaense, Mudanjiang phlebovirus, Gomselga Virus, and Onega tick phlebovirus) in Russia” submitted for publication in PLOS One.

We thank you for valuable suggestions that allowed us to make the manuscript more convincing and understandable. We accepted your suggestion and made corresponding change in the manuscript. We revised our article. Below please find our detailed responses to your questions and comments. All modifications in the manuscript have been highlighted in red.

2. Point-by-point response to Academic Editor Comments

Comment 1: Please ensure that your manuscript meets PLOS ONE’s style requirements, including those for file naming. The PLOS ONE style templates can be found at https://journals.plos.org/plosone/s/file?id=wjVg/PLOSOne_formatting_sample_main_body.pdf and https://journals.plos.org/plosone/s/file?id=ba62/PLOSOne_formatting_sample_title_authors_affiliations.pdf

Response 1: We have carefully reviewed the PLOS ONE style guidelines and have revised our manuscript accordingly to ensure full compliance.

Comment 2: In your Methods section, please provide additional information regarding the permits you obtained for the work. Please ensure you have included the full name of the authority that approved the field site access and, if no permits were required, a brief statement explaining why.

Response 2: We thank the reviewer for raising this important point regarding permits and ethical approvals for field work. Tick collection was performed exclusively from vegetation using the flagging method; no ticks were collected from humans seeking medical care or from animal hosts (mammals or birds). According to the legislation of the Russian Federation, this type of environmental sampling does not require approval from an ethical committee or specific collection permits, as it does not involve vertebrate animals or human subjects. Field work was conducted by qualified specialists from the zoological departments of the Regional Centers for Hygiene and Epidemiology (under the Federal Service for Surveillance on Consumer Rights Protection and Human Wellbeing, Rospotrebnadzor), who hold the necessary certifications and authorizations for entomological surveillance activities in accordance with Russian federal and regional regulations. All collection sites were located on publicly accessible lands or areas where routine epidemiological monitoring is routinely performed by these authorized institutions.

We have added the corresponding clarification to the Methods section of the revised manuscript (lines 119-129) to ensure transparency regarding the regulatory framework governing our field work. We have also added a formal acknowledgment to the Acknowledgments section, expressing our sincere gratitude to the staff of the Regional Centers for Hygiene and Epidemiology for providing the tick specimens used in this study.

Comment 3: Thank you for stating the following financial disclosure: “The work was supported by the Ministry of Science and Higher Education of the Russian Federation (The Federal scientific-technical program for genetic technologies development for 2019-2030, Agreement № 075-15-2025-526).”

If this statement is not correct you must amend it as needed. Please include this amended Role of Funder statement in your cover letter; we will change the online submission form on your behalf.

Response 3: We thank for their attention to the financial disclosure statement. We apologize for the confusion caused by the inadvertent inclusion of a funding acknowledgment that was not applicable to this study. The funders had no role in study design, data collection and analysis, decision to publish, or preparation of the manuscript.

Comment 4: Please note that your Data Availability Statement is currently missing a direct link to access each database. If your manuscript is accepted for publication, you will be asked to provide these details on a very short timeline. We therefore suggest that you provide this information now, though we will not hold up the peer review process if you are unable.

Response 4: We have updated the statement in the revised manuscript to include direct, persistent links to all publicly deposited datasets.

Comment 5: We note that Figure 1 in your submission contain map images which may be copyrighted. All PLOS content is published under the Creative Commons Attribution License (CC BY 4.0), which means that the manuscript, images, and Supporting Information files will be freely available online, and any third party is permitted to access, download, copy, distribute, and use these materials in any way, even commercially, with proper attribution. For these reasons, we cannot publish previously copyrighted maps or satellite images created using proprietary data, such as Google software (Google Maps, Street View, and Earth). For more information, see our copyright guidelines: http://journals.plos.org/plosone/s/licenses-and-copyright.

Response 5: We thank the editorial office for the detailed clarification regarding copyright requirements for map images under the CC BY 4.0. After careful consideration, we have decided to remove Figure 1 from the revised manuscript to ensure full compliance with PLOS ONE’s licensing policy.

Comment 6: Please upload a new copy of Figures 2, 3, 4, 5, and 6 as the detail is not clear. Please follow the link for more information: https://journals.plos.org/plosone/s/figures

Response 6: We have revised Figures to ensure they meet PLOS ONE’s technical and aesthetic requirements.

Comment 7: We note that there is identifying data in the Supporting Information file < S1. Table.pdf >. Due to the inclusion of these potentially identifying data, we have removed this file from your file inventory. Prior to sharing human research participant data, authors should consult with an ethics committee to ensure data are shared in accordance with participant consent and all applicable local laws.

Response 7: We fully agree with the removal of <S1 Table.pdf> from the file inventory due to the presence of potentially identifying information.

Comment 8: If the reviewer comments include a recommendation to cite specific previously published works, please review and evaluate these publications to determine whether they are relevant and should be cited. There is no requirement to cite these works unless the editor has indicated otherwise.

Response 8: We have carefully reviewed all reviewer comments and confirm that none of the reviewers recommended the citation of specific previously published works. All references included in the revised manuscript were selected based on their direct relevance to the study’s methodology, results, and interpretation.

Comment 9: Please review your reference list to ensure that it is complete and correct. If you have cited papers that have been retracted, please include the rationale for doing so in the manuscript text, or remove these references and replace them with relevant current references. Any changes to the reference list should be mentioned in the rebuttal letter that accompanies your revised manuscript. If you need to cite a retracted article, indicate the article’s retracted status in the References list and also include a citation and full reference for the retraction notice.

Response 9: We thank the editor for this important reminder regarding reference list integrity. We have thoroughly reviewed the complete reference list in the revised manuscript to ensure accuracy, completeness, and compliance with PLOS ONE guidelines.

3. Point-by-point response to Reviewer 1 Comments

Reviewer 1: The manuscript is of interest because it provides new information about the circulation of phleboviruses in Russia. Below I present my comments, eliminating which the authors can improve their manuscript

Comments 1: If you describe the segments from which translationally processed proteins are synthesized, you should indicate, including in the figures, the sites of cleavage or other modifications.

When describing the M segment, to understand the biology of the virus, it is more appropriate to look at the structure of the Gc and Gn proteins separately, rather than the precursor. In the table, you also describe the Gc fusion domain, but you don’t provide its structure anywhere. What are its features, and what about the fusion loop? This should be examined in more detail.

Figure 3 requires a clearer description of the structures shown. The lower part of the figure is unclear. In the text you describes RdRp and regions 182, 657, and "in addition," three regions with no currently assigned functional role. However, the figure also includes regions with amino acid length 293 and 800. What is this? These domains are not listed in Table S4. The figure has no description at all for the regions highlighted in gray.

Response 1: Thank you for this insightful comment. See revised Figure 2.

We have completely reorganized the description of the M segment. Instead of the polyprotein, we now describe Gn (formerly G1, 522 aa) and Gc (which comprises the G2 fusion domain, 312 aa, and the G2 C terminal domain, 86 aa) separately. We have added the predicted cleavage sites to the schematic of the M segment. Based on the GenBank annotation of the M segment polyprotein (isolate Tuva-46), we have added the cleavage sites to Figure 2 according to the GenBank annotation of the M polyprotein, the precursor is cleaved: after residue 19, between residues 542 and 544 (Gn–Gc junction), between residues 856 and 881 (within Gc, separating fusion domain and C terminal domain).

The lower part (L segment) is now clearly divided into functional domains with distinct labels. All regions shown in the figure (182 aa endonuclease, 237 aa unknown, 183 aa unknown, 697 aa RdRp, 800 aa unknown) are now listed and described in Figure 2, with their coordinates and predicted pTM/pLDDT values.

We agree that discussing the mature Gn and Gc glycoproteins separately provides better insight into the virus biology, rather than focusing solely on the precursor. We have revised the relevant Figure 2 describe Gn (G1 domain, residues 20–542; pfam07243, CDD:369282) and Gc (Phlebovirus_G2 fusion domain, residues 544–856; pfam07245, CDD:462122, followed by G2 C-terminal domain, residues 881–966; pfam19019, CDD:408789) individually.

Regarding the Gc fusion domain (G2), its key structural features align with conserved phlebovirus architecture: it contains a β-sheet-rich core (as predicted by CDD:462122).

The fusion loop (FL) is a key structural element responsible for membrane fusion in class II fusion proteins. In our Gc protein sequence, the FL is predicted to reside at the tip of domain 2, spanning approximately residues 670 to 684. The amino acid sequence of this putative fusion loop is: QCGGAGCGCFNIHAS. See lines 353-357.

S1 Figure. The putative fusion loop of Gc MKWV Tuva-46 isolate (hydrogen bonds are shown in blue).

S2 Figure. Comparison of the Gc protein from Mukawa virus (MKWV, isolate Tuva-46, colored violet) with the crystal structure of glycoprotein C from Rift Valley fever virus (PDB: 4HJ1, shown in blue).

Comments 2: Further in the text is a description of the nucleoprotein structure analysis. Why was this protein specifically given such a detailed analysis, while others, such as Gc, were not? In Figure 3, it’s also worth separating the top and bottom halves of the figure, labeling them A and B, and providing a separate description for each part.

Response 2: We thank the reviewer for this insightful comment and apologize for not making our rationale clearer in the original manuscript.

The nucleoprotein was selected for detailed structural analysis due to its high evolutionary conservation and critical role in ribonucleoprotein complex assembly. The high confidence of its predicted tertiary structure enabled reliable assessment of the impact of identified amino acid substitutions on functional domains, an approach less applicable to more variable glycoproteins. See lines 371-372.

We have revised Figure 3 accordingly. The figure legend has been updated to provide separate, detailed descriptions for each panel.

4. Point-by-point response to Reviewer 2 Comments

Reviewer 2: This study provides valuable information regarding tick-associated phleboviruses, specifically addressing geographical variability, prevalence rates, and relatedness to other zoonotic potential lineages. The data and supporting information are well-presented, and the scientific findings warrant publication. The Introduction is exceptionally well-written, offering a clear historical context for phlebovirus in Asia/Russia and surrounding regions. Below are minor comments need the author’s attention.

Methodology:

Comments 1: Lines 128-129: Please provide more detail on the tick sample homogenization process. Given the hard exoskeleton, a physical force step is necessary to initially break open the tick before further processing.

Response 1: We thank the reviewer for this important observation regarding the tick sample homogenization process. We agree that providing detailed information on the physical disruption step is crucial for reproducibility, given the hard exoskeleton of the ticks. We have revised the Methods section (Lines 137–142) to include specific details.

«Tick samples were mechanically homogenized in 300 µL of sterile physiological saline using a TissueLyser LT homogenizer (Qiagen, Hilden, Germany). This instrument operates by high-speed shaking of sample tubes in the presence of solid particles, ensuring rapid and uniform disruption of the hard exoskeleton. Two steel beads (4 mm diameter; Servicebio, Wuhan, China) were added to each tube containing an individual tick specimen. Homogenization was performed at a shaking frequency of 50 Hz (50 shakes/sec) for 5 minutes»

Comments 2: Lines 132-138: RNA stability is more fragile than DNA. The successful PCR amplification of DNA target does not guarantee the integrity of the original RNA template. This point requires clarification.

Response 2: We thank the reviewer for this important observation regarding the differential stability of RNA and DNA. We fully acknowledge that successful amplification of a DNA target (COI) does not strictly guarantee the integrity of the original RNA template.

To mitigate the risk of RNA degradation, reverse transcription was performed immediately following total nucleic acid extraction. The COI PCR assay served primarily as a control for extraction efficiency and the absence of PCR inhibitors; samples failing this control would have been excluded to prevent false-negative viral results. In our study, all samples tested positive for the COI marker. Additionally, this procedure verified the morphological identification of the tick species. However, we agree with the reviewer that it remains possible that some samples contained intact tick DNA while the viral RNA was degraded prior to extraction. We have added a statement to the Methods section (lines 147-149 and 152-153) to clarify this limitation and the steps taken to minimize it.

Results/Figures/Tables:

Comments 3: Table 1: The percentage of infection is missing for the Mukawa virus under the Chita region.

Response 3: We thank the reviewer for reviewing Table 1 and for pointing out this omission. We have corrected the table to include the missing prevalence value for Mukawa virus in the Chita region.

Comments 4: Figures: The quality of the pictures need improvement, particularly the clarity of the dots representing each detected virus (Fig 1).

Response 4: We would like to kindly clarify that Figure 1 has been removed from the revised manuscript following a separate request from the PLOS ONE editorial office concerning copyright compliance for map images under the CC BY 4.0.

General:

Comments 5: Please thoroughly check the entire manuscript for typos and correct all mistakes. Examples of lines needing attention include 316 and 371.

Response 5: W

---

## [Editor Report · Decision Letter 1]

3 May 2026

Expanding diversity of tick-borne phleboviruses (Phlebovirus mukawaense, Mudanjiang phlebovirus, Gomselga Virus, and Onega tick phlebovirus) in Russia

PONE-D-26-10722R1

Dear Dr. Gladysheva,

We’re pleased to inform you that your manuscript has been judged scientifically suitable for publication and will be formally accepted for publication once it meets all outstanding technical requirements.

An invoice will be generated when your article is formally accepted. Please note, if your institution has a publishing partnership with PLOS and your article meets the relevant criteria, all or part of your publication costs will be covered. Please make sure your user information is up-to-date by logging into Editorial Manager at Editorial Manager® and clicking the ‘Update My Information’ link at the top of the page. For questions related to billing, please contact billing support.

Kind regards,

Humberto Julio Debat

Academic Editor

PLOS One

Additional Editor Comments (optional):

Reviewers’ comments:

---

## [Editor Report · Acceptance letter]

PONE-D-26-10722R1

PLOS One

Dear Dr. Gladysheva,

I’m pleased to inform you that your manuscript has been deemed suitable for publication in PLOS One. Congratulations! Your manuscript is now being handed over to our production team.

Lastly, if your institution or institutions have a press office, please let them know about your upcoming paper now to help maximize its impact. If they’ll be preparing press materials, please inform our press team within the next 48 hours. Your manuscript will remain under strict press embargo until 2 pm Eastern Time on the date of publication. For more information, please contact onepress@plos.org.

Kind regards,

on behalf of

Professor Humberto Julio Debat

Academic Editor

PLOS One